# Notch-RBPJ Pathway for the Differentiation of Bone Marrow Mesenchymal Stem Cells in Femoral Head Necrosis

**DOI:** 10.3390/ijms24076295

**Published:** 2023-03-27

**Authors:** Kangping Liu, Hongfan Ge, Chengyin Liu, Yixin Jiang, Yaling Yu, Zhenlei Zhou

**Affiliations:** Department of Veterinary Clinical Science, College of Veterinary Medicine, Nanjing Agricultural University, Nanjing 210095, China

**Keywords:** femoral head necrosis, BMSCs, Notch-RBPJ, differentiation, osteogenic, chondrogenesis

## Abstract

Femoral head necrosis (FHN) is a common leg disease in broilers, resulting in economic losses in the poultry industry. The occurrence of FHN is closely related to the decrease in the number of bone marrow mesenchymal stem cells (BMSCs) and the change in differentiation direction. This study aimed to investigate the function of differentiation of BMSCs in the development of FHN. We isolated and cultured BMSCs from spontaneous FHN-affected broilers and normal broilers, assessed the ability of BMSCs into three lineages by multiple staining methods, and found that BMSCs isolated from FHN-affected broilers demonstrated enhanced lipogenic differentiation, activated Notch-RBPJ signaling pathway, and diminished osteogenic and chondrogenic differentiation. The treatment of BMSCs with methylprednisolone (MP) revealed a significant decrease in the expressions of Runx2, BMP2, Col2a1 and Aggrecan, while the expressions of p-Notch1/Notch1, Notch2 and RBPJ were increased significantly. Jagged-1 (JAG-1, Notch activator)/DAPT (γ-secretase inhibitor) could promote/inhibit the osteogenic or chondrogenic ability of MP-treated BMSCs, respectively, whereas the differentiation ability of BMSCs was restored after transfection with si-RBPJ. The above results suggest that the Notch-RBPJ pathway plays important role in FHN progression by modulating the osteogenic and chondrogenic differentiation of BMSCs.

## 1. Introduction

Bone marrow mesenchymal stem cells (BMSCs) are cells present in bone marrow tissue with no hematopoietic potential, easy access, and demonstrate multiple differentiation potentials and the ability to secrete trophic factors [1,2]. Over the past 30 years, BMSCs have been widely studied for their clinical applications in tissue engineering and regenerative medicine, such as bone and cartilage reconstruction and wound healing [3]. Related researches report that BMSCs play a vital role in osteonecrosis repair, which invades the necrotic area, differentiates, and forms new bone to complete the repair [4,5]. Femoral head necrosis (FHN) is a disease of abnormal bone metabolism, and its pathogenesis is closely associated with a decrease in the activity and number of BMSCs [6,7,8]. In addition, the osteogenic capacity of MSCs derived from patients with steroid-induced FHN was compromised, with increased adipogenesis and decreased proliferation rates [9,10,11,12,13]. However, the mechanism in the differentiation alteration of BMSCs in FHN remains poorly understood.

BMSCs are regulated by several signaling pathways, among which the Notch signaling pathway is involved in regulating the proliferation and differentiation of BMSCs cells, determining cell fate and participating in the development of embryonic and adult tissues, such as skeletal tissues [14,15,16,17]. Under the action of external factors, the Notch receptor-ligand junctions at the cell surface are decoded, and the receptor protein is cleaved by the metalloproteinase tumor necrosis factor-α converting enzyme (TACE) and γ-secretase complex, releasing the Notch intracellular structural domain (NICD) [3,18,19]. NICD enters the nucleus and binds to the transcription factor RBPJ to form a transcriptional complex that initiates the expression of downstream genes such as hairy and enhancer of split (HES) and hairy/enhancer-of-split related (HEY) to exert biological effects [18]. Dong et al. demonstrated that JAG 1-mediated Notch signaling promoted the maintenance and expansion of BMSCs while enhancing their osteogenic differentiation capacity [20]. It reported that circ-DAB1 furtherance cell multiplication and osteogenic differentiation of BMSCs through the Notch-RBPJ pathway [21]. These studies highlight the need for the Notch signaling pathway in the osteogenic differentiation of BMSCs.

Therefore, the aim of this study was to compare and analyze the changes in BMSCs differentiation in spontaneous FHN-affected broilers, and simultaneously conduct the in vitro cell assay to investigate the related mechanisms of the Notch-RBPJ signaling pathway in BMSCs differentiation progression.

## 2. Results

### 2.1. Decreased Osteogenic and Chondrogenic Differentiation and Increased Lipogenic Capacity of BMSCs in Spontaneous FHN Broiler Femur Tissue

Mesenchymal stem cells (MSCs) applied in this study were isolated from bone marrow in broilers and were confirmed for their identity and suitability for usage according to current criteria (Figure 1A,B). To investigate the effective regulator of the Notch pathway on osteogenesis and chondrogenesis in FHN-affected broilers, femoral head tissue samples and BMSCs were collected. The results of ARS staining and alcian blue staining were shown in Figure 1B focusing on the differentiation ability of BMSCs. BMSCs in the control group exhibited normal osteogenic chondrogenic and adipogenesis differentiation, while attenuated osteogenic and chondrogenic differentiation and increased lipogenic capacity were observed in FHN-affected broilers, indicating the osteogenic and chondrogenic differentiate dysfunction of the BMSCs. Meanwhile, the levels of chondrogenic and osteogenic metabolism-related proteins in femoral head tissues of FHN-affected broilers were changed as previously described, that was, the levels of Col2a1, Aggrecan and BMP2 proteins were significantly downregulated, while the protein levels of matrix metalloproteinase hydrolase (MMP13) were significantly elevated (Figure 1C,D). A significantly upregulated transcript level and protein expression levels of Notch pathway-related genes occurred in FHN-affected broilers (Figure 1E,F), indicating that the Notch pathway was activated.

### 2.2. MP Causes the Homeostatic Imbalance of BMSCs and Activation of the Notch Pathway

BMSCs were exposed to various doses of MP (200, 400, 800, and 1200 µg/mL) for 6, 12, 24 and 48 h, respectively, to evaluate the cytotoxicity of MP. From the result of the CCK-8 assay (Figure 2A) and the apoptosis related genes transcription result in BMSCs (Figure 2B), it was appropriate that BMSCs were treated with 400 µg/mL MP for 24 h (cell viability = 70.4252%) in the present study. Treatment with MP reduced osteogenesis and chondrogenic differentiation but enhanced lipogenic capacity in BMSCs (Figure 2C). Chondrogenesis and osteogenesis metabolism-related genes, such as Col2a1, Aggrecan, BMP2 and Runx2, were determined using qRT-PCR showing that they were downregulated with MP treatment, whereas the mRNA levels of MMP9 were extremely increased inversely (Figure 2D,F). Moreover, the protein levels of Col2a1, Aggrecan, BMP2 and Runx2 were decreased (Figure 2E,G). Simultaneously, genes and protein levels related to the Notch pathway were changed identically to those of FHN-affected broilers, specifically, the mRNA level of Notch1, Notch2, JAG-1, JAG-2, RBPJ and Hes1 were increased, and the protein level of Notch2 and RBPJ was significantly promoted in MP treatment compared to the control group (Figure 2H,I). The immunofluorescence assay results were concordant with the western blotting assay, indicating that RBPJ expression was extremely increased in MP-treated BMSCs compared with the control (Figure 2J). The results (Figure 2) indicate that MP induces a homeostatic imbalance in BMSCs and activates the Notch pathway.

### 2.3. Regulation of MP-Induced Cellular Osteogenesis and Chondrogenesis Metabolism by the Notch Pathway

To investigate the role of the Notch pathway in regulating the differentiation ability of BMSCs, Jagged-1 (JAG-1, Notch agonist) and DAPT (γ-secretase inhibitor) were used in this study. The concentration of BMSCs treated with JAG-1 and DAPT were determined to be 6.0 μg/mL (Figure 3A), and 2.0 μM (Figure 4A) exposure for 24 h, respectively. Firstly, Notch ligands (JAG-1, JAG-2 and DLL-1) and RBPJ were determined using qRT-PCR showing that it was significantly upregulated with JAG-1 treatment compared to the control group, whereas the mRNA level of Hes1 was extremely significant increased (Figure 3B). Moreover, the protein levels of pNotch1/Notch1 and RBPJ were increased in the JAG-1 group compared to the control group (Figure 3C). The qRT-PCR test revealed a significant decrease in Aggrecan of JAG-1 group (Figure 3D,F). Protein expressions of Aggrecan and Runx2 in the MP+JAG-1 treatment was significantly decreased compared to the control treatment, while the level of MMP9 and MMP13 were elevated inversely (Figure 3E,G). Overall, JAG-1 activates the Notch pathway of BMSCs and aggravates the reduction in BMSCs’ osteogenic and chondrogenic metabolic capacity caused by MP.

DAPT acts as an inhibitor of γ-secretase, thus playing the role of restraining Notch activation. Exposure to DAPT was able to inhibit the mRNA level of Notch1 in BMSCs compared to controls’ (Figure 4B). The protein level of pNotch1/Notch1 was extremely promoted when exposure to MP+DAPT compared to the MP-treated BMSCs (Figure 4C). The mRNA level of Col2a1, Aggrecan, BMP2 and Runx2 were tested by qRT-PCR exhibiting that these were significantly elevated with MP+DAPT group compared to the control group (Figure 4D,F). Nevertheless, the mRNA levels of MMP9 and MMP13 were significantly decreased (Figure 4F). The qRT-PCR assay results of BMP2 and MMP13 were confirmed by the western blotting data (Figure 4E,G). These findings suggested that the Notch pathway had the capacity to regulate the differentiation of BMSCs induced by MP.

### 2.4. MP-Induced Alterations in BMSCs Differentiation Are Mitigated by si-RBPJ

The transfection of si-RBPJ or negative control (NC) was performed with or without MP exposure. As demonstrated in Figure 5A, transfection of si-RBPJ effectively reduced expression to 30.38%. The protein level of RBPJ was downregulated by si-RBPJ compared to the NC group (Figure 5B). The immunofluorescence result confirmed the effect of si-RBPJ transfection on RBPJ level (Figure 5G). The effect of RBPJ on BMSCs cartilage metabolism was manifested as si-RBPJ significantly increased the decrease in the expression of Col2a1 caused by MP, with both mRNA level and protein level (Figure 5C,D). The changes in BMP2 and MMP13 in BMSCs induced by MP were significantly improved by si-RBPJ (Figure 5E,F). Overall, the transformations of BMSCs osteogenesis and chondrogenesis metabolism induced by MP can be regulated by RBPJ.

## 3. Discussion

As a bone disorder, FHN is characterized by periarticular bone disintegration and cartilage degeneration [22]. According to our previous study, FHN-affected broilers showed lipid metabolism disorders, increased lipid droplet formation, an imbalance in cartilage homeostasis, reduced extracellular matrix (ECM) formation, and simultaneous apoptosis [23,24,25]. Reduced osteogenic and chondrogenic differentiation of BMSCs in FHN-affected broilers and enhanced lipogenic differentiation were observed in the present study (Figure 1B). The decrease in the number of osteoblasts and cartilage degeneration is one of the main reasons for the occurrence of FHN [26]. Therefore, in this study, we further investigated the signaling pathway in osteogenic and chondrogenic differentiation of BMSCs, and analyzed the connection between BMSCs differentiation and the occurrence of FHN.

Runx2 is a transcription factor that modulates the osteopontin (OPN), bone matrix genes osteocalcin (OCN) and bone sialoprotein (BSP) of mesenchymal stem cells [27], and sustains the presentation of OPN and BSP during the osteoblast differentiation [28]. Related studies showed that BMP2 has a strong osteogenic capacity, which can greatly increase OCN expression [29] and is a necessary and sufficient factor for the irreversible induction of bone formation [30]. Matrices metallopeptidase 9 (MMP9) and 13 (MMP13) are capable of degrading multiple extracellular matrix proteins [31]. Xu et al. reported that the femoral head cartilage in patients with early FHN demonstrated early cartilage degeneration while the expressions of Aggrecan and Col2a1 were decreased [32]. As shown in Figure 1, the ARS staining and alcian blue staining of BMSCs in spontaneous FHN-affected broilers showed that the number of osteoblasts and chondrocytes was significantly reduced compared with normal broilers, and the protein levels of osteogenesis and chondrogenic-related factors BMP2, Col2a1 and Aggrecan were decreased significantly, while the protein level of MMP13 was significantly promoted, indicating that the osteogenesis and chondrogenic differentiation of BMSCs in FHN-affected broilers were significantly reduced.

MP is commonly used to induce the establishment of FHN cell models, so BMSCs was exposed to MP for in vitro studies. Notch signaling is a highly conserved pathway that takes place via direct correlation between Notch receptors and ligands on neighboring cells [33]. According to our work, the Notch of BMSCs derived from spontaneous FHN broilers was activated, while the same result was found in vitro MP model (Figure 1 and Figure 2), which is consistent with the regulatory effect of MP/dexamethasone on human MSC [10,34]. At present, there are two views on the regulation of Notch signal on the differentiation of MSCs. On the one hand, Notch signal transduction promotes the mineralization of human bone-derived cells [35,36], on the other hand, it is believed that Notch RBP-Jk has a negative effect on BMP-induced MSCs differentiation [37,38,39]. Notch signaling affects bone remodeling and regeneration processes after fracture [40,41]. Notch receptors are upregulated during fracture healing and their pharmacological inhibition contributes to fracture healing [40], and the Notch ligand JAG-1 has a similar effect [41]. The addition of JAG-1 in the present study did aggravate the weakened osteogenesis of BMSCs induced by MP (Figure 3F,G). The present study’s findings are consistent with Oldershaw et al., showing that sustained activation of the Notch pathway in BMSCs caused by JAG-1 treatment inhibited chondrogenic differentiation (Figure 3D,E) [42]. Notch signaling is obstructed by γ-secretase inhibitor (DAPT), thereby inhibiting the expression of downstream target genes and repressing cell growth and proliferation [43,44]. In Figure 4, by inhibiting γ-secretase activity, DAPT was able to attenuate the Notch activation induced by MP treatment and alleviate the MP-induced decrease in osteogenesis and chondrogenesis metabolism in BMSCs. These data (Figure 1, Figure 2, Figure 3 and Figure 4) reveal that the Notch pathway is involved in the regulation of osteogenesis and chondrogenic differentiation of BMSCs.

The activation of the Notch pathway triggers the release of NICD from the cell membrane and converts RBPJ from blocking to transcription of Notch target genes by supplementing additional coactivators upon nuclear transposition [45]. In this work, transfection with si-RBPJ attenuated the results of MP-induced reduction of bone formation and cartilage metabolism in BMSCs, which was consistent with exposure to DAPT, indicating that the protein and mRNA levels of Col2a1 were significantly increased with si-RBPJ compared with the MP treatment group (Figure 4 and Figure 5). Thus, we would predict that Notch-RBPJ modulates the osteogenic and chondrogenic differentiation of BMSCs in FHN progression.

In conclusion, we found that the Notch-RBPJ pathway was activated in FHN-affected broilers, suppressing the osteogenic and chondrogenic differentiation capacity of BMSCs (Figure 6).

## 4. Materials and Methods

### 4.1. Sample Collection

The spontaneous FHN and healthy broiler BMSCs were cultured separately at 21 days of age, and the difference in osteogenic, chondrogenic and lipogenic differentiation ability was examined to demonstrate that the occurrence of FHN is related to the change in the direction of BMSCs differentiation. Notch, as one of the most conserved pathways, has been studied more frequently on the regulation of stem cell differentiation, but there are few reports on the regulatory role of Notch in regulating the direction of BMSCs differentiation on the occurrence of FHN. The regulatory role of Notch in regulating the direction of BMSCs differentiation on FHN development has been rarely reported. BMSCs were isolated and cultured from healthy broilers at 10–21 days of age for in vitro experimental studies. Both sexes’ broilers (Ross 308) were acquired from a farm in Jiangsu Province. Femoral tissue from normal broilers and spontaneous FHN broilers was collected and stored in liquid nitrogen, as previously reported [23,24]. All animal protocols were approved by the Animal Protection and Use Committee of Nanjing Agricultural University (#NJAU-Poult-2019102205, approved on 22 October 2019).

### 4.2. Isolation, Culture, and Identification of Primary BMSCs

BMSCs were harvested from the bone marrow of 21-day-old normal and spontaneous FHN broilers. That is, BMSCs were rinsed and collected from broiler femurs and tibias, placed in T25 flasks, and incubated overnight at 37 °C in an incubator with 5% CO_2_. Then, non-adherent cells were removed by rinsing twice with phosphate-buffered saline (PBS, Basalmedia Biotechnology Co., Ltd., Shanghai, China). The culture medium was changed every 2 d. After 10–12 d of culture, when the cell density reached 80%, passaging culture was performed. After conventional cell passaging culture (0.25% trypsin/EDTA (Solarbio Biotechnology Co., Ltd., Beijing, China) digestion at 37 °C, 2 min), the cell suspension was collected, centrifuged at 1000 rpm for 5 min, the supernatant was discarded, and the complete medium was resuspended and inoculated into T25 culture flasks for further culture, recorded as P1. To verify the BMSCs, P3 cells were subjected to morphological observations, osteogenic and adipogenic induction, and phenotypic analyses.

### 4.3. Cell Treatment

The medium was replaced with methylprednisolone (MP, Sinopharm, Jiaozuo, China) for 24 h to explore the effect of MP on BMSCs (P3–P5). The cells were handled with MP, DAPT (MCE, Shanghai, China), JAG-1 (Solarbio Biotechnology Co., Ltd., Beijing, China) and si-RBPJ (Gene Pharma, Shanghai, China) to observe the influence of Notch on MP-induced cell differentiation.

### 4.4. Cell Viability Assay

In brief, the BMSCs from normal broilers were plated into 96-well plates. After treatment of BMSCs with MP, DAPT or JAG-1, cells were incubated with CCK-8 (Vazyme biotech Co., Ltd., Nanjing, China) in accordance with the instructions. The absorbance at 490 nm was measured using a microplate reader.

### 4.5. Osteogenesis Differentiation

For osteogenesis, BMSCs (P3, 4 × 105 cells/cm^2^) were cultured in DMEM/F12 (Basalmedia Biotechnology Co., Ltd., Shanghai, China) plus 10% FBS (Gibco, USA) added with 5 μg/mL ascorbate, 10 mM β-glycerol phosphate and 10 nM dexamethasone (Solarbio Biotechnology Co., Ltd., Beijing, China). The culture medium was replaced every two days.

### 4.6. Lipogenic Differentiation

For adipogenesis, BMSCs (P3, 5 × 105 cells/cm^2^) were cultured DMEM/F12 plus 10% FBS added with 0.5 mM isobutylmethylxanthine (IBMX, Beyotime Biotechnology, Shanghai, China), 1.0 μM dexamethasone, 10 μg/mL insulin and 0.1 mM indomethacin (Solarbio Biotechnology Co., Ltd., Beijing, China). The culture medium was replaced every two days.

### 4.7. Chondrogenic Differentiation

To promote chondrogenic differentiation, BMSCs (P3, 5 × 105 cells/cm^2^) were cultured in DMEM/F12 supplemented with 100 ng/mL TGF-β1 (Novoprotein, Suzhou, China), 10 nM dexamethasone, 50 μg/mL ascorbic acid, 6.25 μg/mL bovine insulin and transferrin, and 1.25 μg/mL bovine serum albumin (Solarbio Biotechnology Co., Ltd., Beijing, China), and the medium was changed every two days.

### 4.8. Alizarin Red S (ARS) Staining

After 21 d of osteogenic induction culture, cells were fixed with 4% paraformaldehyde (PFA, Servicebio Biotechnology Co., Ltd., Wuhan, China) and stained with alizarin red S staining (Solarbio Biotechnology Co., Ltd., Beijing, China) solution for 30 min, and neutral resin was used to seal the slices for observation using Nikon fully automatic orthomosaic microscope (Nikon, Tokyo, Japan).

### 4.9. Oil Red O Staining

After 21 d of lipid-forming induction culture, cells were fixed with 4% PFA and stained with 0.3% Oil Red O staining (Solarbio Biotechnology Co., Ltd., Beijing, China) solution for 30 min, and neutral resin was used to seal the slices for observation using Nikon fully automatic orthomosaic microscope (Nikon, Tokyo, Japan).

### 4.10. Alcian Blue Staining

After 21 d of chondrogenesis induction, cells were fixed with 4% PFA, soaked in alcian acidified solution for 3 min, then stained with alcian staining solution for 30 min, rinsed, and re-stained into nuclear solid red staining solution for 5 min, and sealed with neutral resin for observation using Nikon fully automatic orthomosaic microscope (Nikon, Tokyo, Japan).

### 4.11. Flow Cytometry Analysis of the Surface Markers

BMSCs of P1 were collected and suspended in PBS supplemented with 4% PFA at a concentration of 1 × 106 cells/mL. Anti-rabbit CD29 antibody (AF0207, Beyotime Biotechnology, Shanghai, China), anti-rabbit CD44 antibody (AF1858, Beyotime Biotechnology, Shanghai, China), anti-rabbit CD34 antibody (AF0102, Beyotime Biotechnology, Shanghai, China) and anti-rabbit HLA-DR (AF2065, Beyotime Biotechnology, Shanghai, China) antibody were added to BMSCs at a concentration of 1:50 and incubated for 30 min at room temperature (RT). After being blocked with 5% BSA for 15 min at RT, goat anti-rabbit fluorescent secondary antibody (Affinity biosciences, Jiangsu, China) was added and incubated for 40 min at RT. PBS was used as the blank. The cells were washed with 2% BSA-PBS and resuspended for detection using BD FACS Verse TM 273 Flow Cytometer (BD Biosciences, NJ, USA).

### 4.12. RNA Isolation and qRT-RCR

Total RNA was extracted from BMSCs, using Trizol and reverse transcribed using HiScript II QRT SuperMix (Vazyme, Nanjing, China), according to the manufacturer’s instruction. RT–qPCR was performed using SYBR Green PCR technology (Vazyme, Nanjing, China) and analysis was performed employing the 2−ΔΔCt method (n = 3). Specific forward and reverse primers used in this study are shown in Table 1. The housekeeping gene GAPDH was used as an internal standard for cDNA quantity and quality.

### 4.13. Western Blot

After treatment and detection of total protein concentration with the BCA Protein Assay Kit (Epizyme Bio, Shanghai, China), total proteins were collected from cells for protein blotting experiments [22,46]. Anti-Notch1 (1:300, Wanlei Bio, Shenyang, China), anti-p-Notch1 (1:300, Wanlei Bio, China), anti-Notch2 (1:300, Wanlei Bio, China), anti-RBPJ (1:1000, Beyotime Biotechnology, Shanghai, China), anti-Runx2 (1:300, Wanlei Bio, China), anti-Col2a1 (1:1000, Beyotime Biotechnology, Shanghai, China), anti-BMP2 (1:300, Wanlei Bio, China), anti-MMP9 (1:300, Wanlei Bio, China), anti-MMP13 (1:300, Wanlei Bio, China), anti-GAPDH (1:1000, Proteintech, Wuhan, China) primary antibodies were used.

### 4.14. Immunofluorescence

Cells were inoculated on sterile crawlers placed in 24-well plates and performed as described previously when 80% confluence was reached [47]. Cells were incubated with anti-RBPJ (1:200, Beyotime Biotechnology, Shanghai, China) overnight at 4 °C. The cells were washed and incubated with the fluorescent secondary antibody for 1 h at RT, then washed with PBST followed by DAPI staining for 5 min. After processing, the images were analyzed using an LSM 710 confocal laser microscope device (Zeiss, Oberkochen, Germany).

### 4.15. Cell Transfection

The si-RBPJ (sense, 5′-AUUUCACUCCAAAUUUACGTT-3′, antisense, 5′- UAAAUUUGGAGUGAAAUUCUG-3′) were chemically synthesized to interfere with RBPJ expression (GenePharma, Shanghai, China). The experimental groups included four groups: (1) negative control (NC) transfection; (2) MP exposure after negative control transfection; (3) si-RBPJ expression; and (4) si-RBPJ expression followed by MP exposure. Cells were transfected according to the instructions of RFect Transfection Reagent (Cat No. 11014, BIOG, China). After 48 h of transfection, the cells were used for further experiments.

### 4.16. Statistics Analysis

The statistical analysis was conducted using SPSS 25.0. A two-tailed Student’s *t*-test was applied when two groups were compared. One-way ANOVA, Duncan and Tukey’s post hoc test was performed to determine the differences among multiple groups. Three times repeating experiments are required. Data are shown as the mean ± SD. For all analyses, *p* < 0.05 were considered statistically significant and *p* < 0.01 means extremely significant.

## Figures and Tables

**Figure 1 ijms-24-06295-f001:**
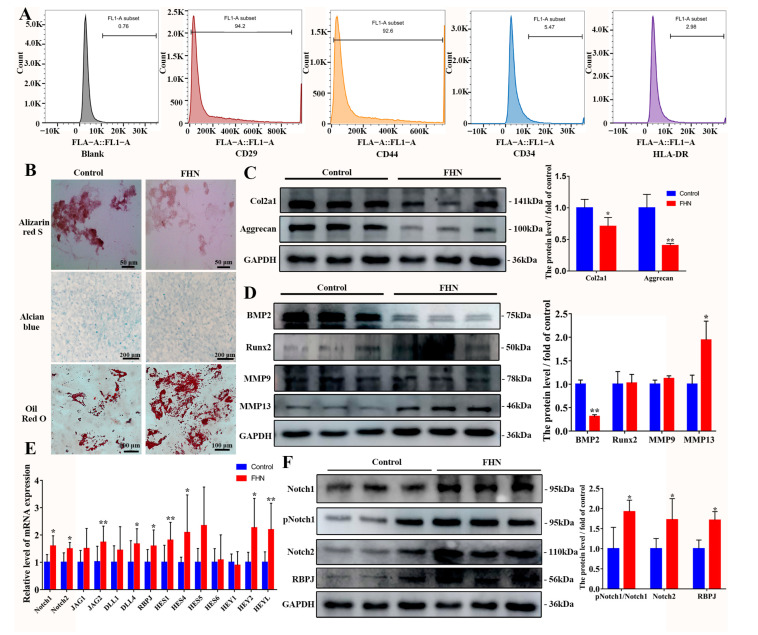
Attenuated osteogenic and chondrogenic differentiation capacity of BMSCs in femur tissues in FHN-affected broiler. (**A**) Identification of BMSCs surface markers by flow cytometry. Positive rate: blank: 0.76%, CD29^+^: 94.2%, CD44^+^: 92.6%; CD34^+^: 5.47%, HLA-DR^+^: 2.98%. (**B**) Osteogenic, chondrogenic and adipogenesis differentiation of BMSCs isolated from normal and FHN-affected broilers in 21 days. The level of bone metabolism and cartilage metabolism-related proteins (**C**,**D**), Notch-related genes (**E**) and proteins (**F**) in femoral head tissues of normal and FHN-affected broilers in 21 days. Scale bar = 50 μm, 100 μm, 200 μm. Compared to the control group, * *p* < 0.05 were considered statistically significant and ** *p* < 0.01 means extremely significant.

**Figure 2 ijms-24-06295-f002:**
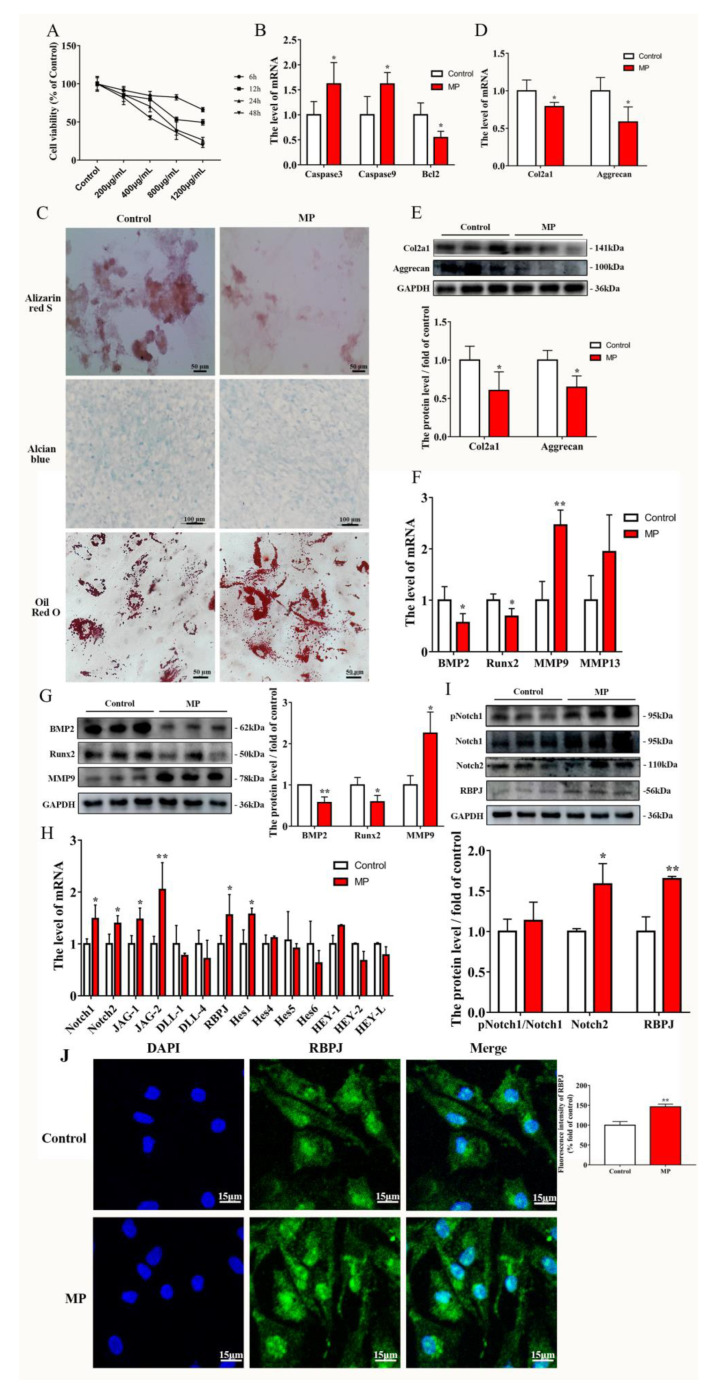
MP exposure activated the notch pathway in BMSCs. (**A**) The cell viability of BMSCs treated with 200, 400, 800, and 1200 µg/mL for 6, 12, 24 and 48 h. (**B**) MP treatment of BMSCs changed the transcript levels of apoptosis-related genes. (**C**) Effect of MP on the level of osteogenesis, chondrogenic and adipogenesis differentiation in BMSCs. Effects of MP on the expression levels of genes (**D**,**F**) and proteins (**E**,**G**) related to bone metabolism and cartilage metabolism in BMSCs. Effect of MP on Notch-related genes (**H**) and protein (**I**) expression levels in BMSCs. (**J**) RBPJ expression stained with Alexa Fluor 488 fluorescence (green) was visualized by immuno-fluorescence, and nuclei were stained with DAPI (blue). Scale bar = 15 µm, 50 µm, 100 µm, 200 µm. Compared to the control group, * *p* < 0.05 were considered statistically significant and ** *p* < 0.01 means extremely significant.

**Figure 3 ijms-24-06295-f003:**
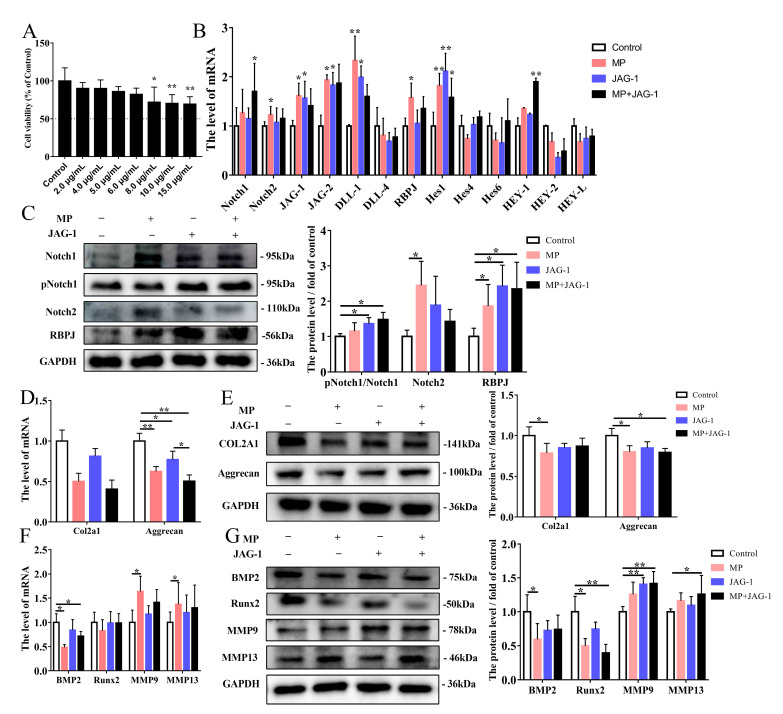
Regulation of MP-induced cell differentiation by JAG-1 activation of the Notch pathway. (**A**) Cell viability under different JAG-1 concentrations in 24 h. (**B**) Effect of JAG-1 on Notch-related genes expression levels in BMSCs. (**C**) The protein expressions of p-Notch1, Notch1, Notch2 and RBPJ were detected by western blotting analysis. (**D**–**E**) The expression levels of Col2a1 and Aggrecan in BMSCs exposed to MP and JAG-1 were determined by qRT-PCR and western blot. (**F**–**G**) Effects of JAG-1 on the mRNA and protein levels of BMP2, Runx2, MMP9 and MMP13 in BMSCs tested by qRT-PCR and western blot. * *p* < 0.05 were considered statistically significant and ** *p* < 0.01 means extremely significant.

**Figure 4 ijms-24-06295-f004:**
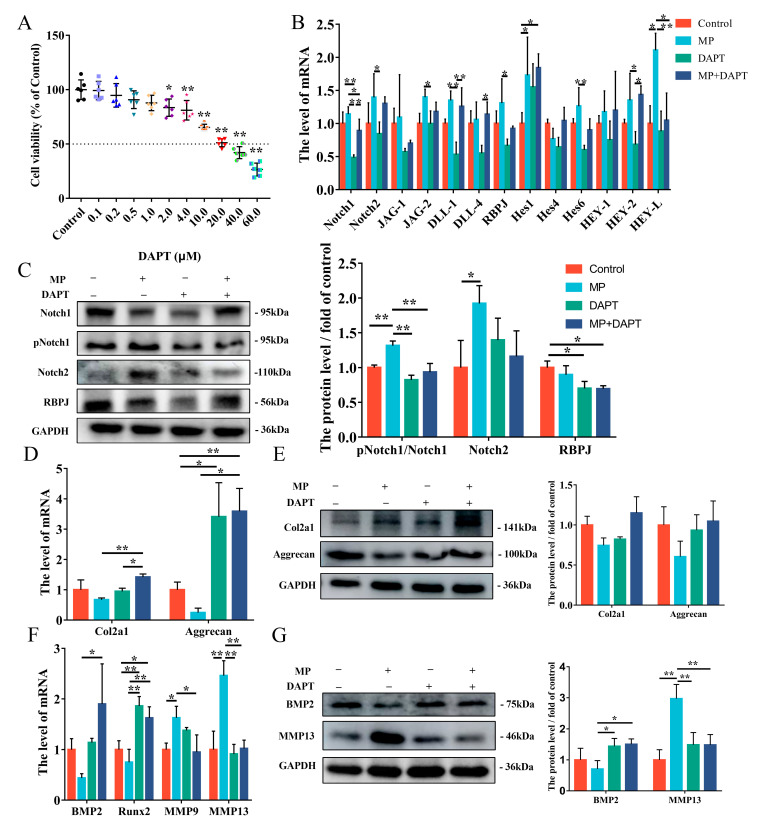
DAPT regulates MP-induced cell differentiation by inhibiting the Notch pathway. (**A**) Cell viability under different DAPT concentrations in 24 h. (**B**) The mRNA level of the Notch pathway exposed to the MP and DAPT in BMSCs. (**C**) The protein expression of p-Notch1, Notch1, Notch2 and RBPJ was detected by western blotting analysis treated with MP and DAPT. (**D**–**E**) The expression levels of Col2a1 and Aggrecan in BMSCs were determined by qRT-PCR and western blot. (**F**) Effects of DAPT on the expression levels of BMP2, Runx2, MMP9 and MMP13 in BMSCs tested by qRT-PCR. (**G**) The protein levels of BMP2 and MMP13 in BMSCs. * *p* < 0.05 were considered statistically significant and ** *p* < 0.01 means extremely significant.

**Figure 5 ijms-24-06295-f005:**
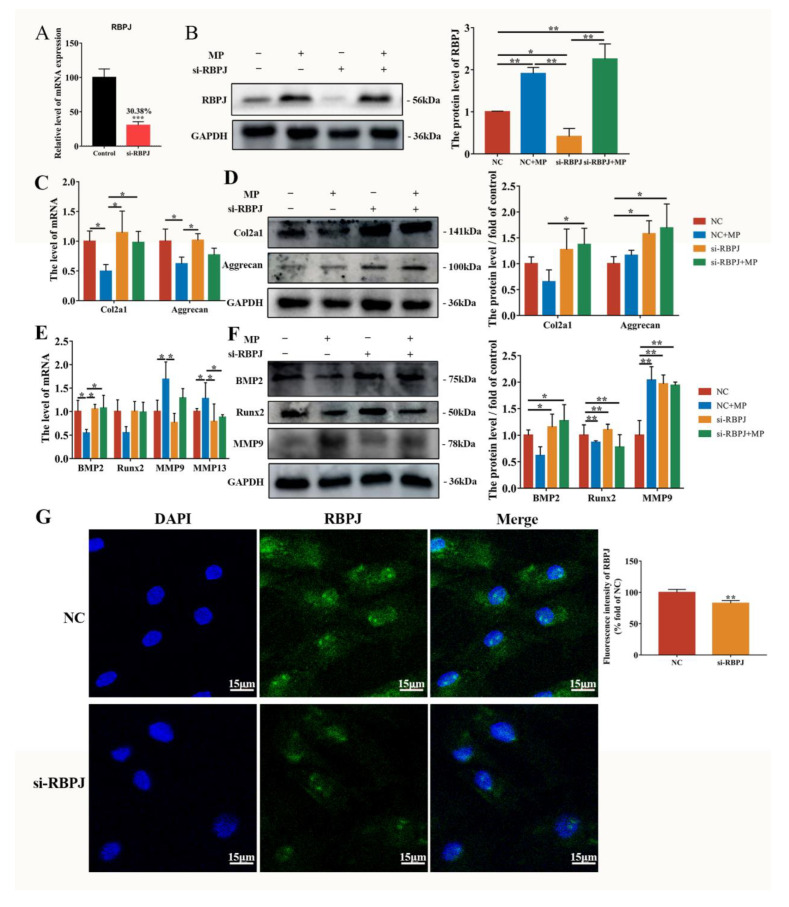
MP-induced alterations in BMSCs differentiation are mitigated by si-RBPJ. (**A**) The transfection efficiency of si-RBPJ; (**B**) effect of si-RBPJ on RBPJ protein expression level in BMSCs. (**C**,**D**) Effects of si-RBPJ on the expression levels of Col2a1 and Aggrecan genes (**C**,**E**) and proteins (**D**,**F**) related to bone metabolism and cartilage metabolism in BMSCs. (**G**) RBPJ expression stained with Alexa Fluor 488 fluorescence (green) was visualized by immuno-fluorescence, and nuclei were stained with DAPI (blue). Scale bar = 15 µm. * *p* < 0.05 were considered statistically significant and ** *p* < 0.01 means extremely significant.

**Figure 6 ijms-24-06295-f006:**
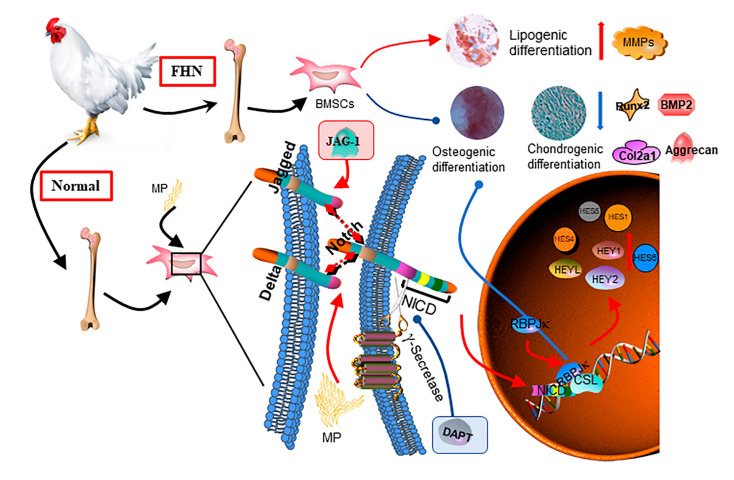
Schematic representation of the regulation of osteogenesis and chondrogenesis metabolism by Notch-RBPJ in BMSCs.

**Table 1 ijms-24-06295-t001:** Sequences of primers used to amplify specific mRNAs by qRT-PCR.

Target Gene	Primer Sequence (5’-3’)
Aggrecan	F: TGCAAGGCAAAGTCTTCTACG
	R: GGCAGGGTTCAGGTAAACG
Col2a1	F: ACCTACAGCGTCTTGGAGGA
	R: ATATCCACGCCAAACTCCTG
BMP2	F: CTCAGCTCAGGCCGTTGTTA
	R: GTCATTCCACCCCACGTCAT
Runx2	F: TTCACAAGCATTTCATCCCTC
	R: TTGCGGACATACCCAGTGACA
MMP9	F: GCCATCACTGAGATCAATGGAG
	R: GATAGAGAAGGCGCCCTGAGT
MMP13	F: AGAGACCCTGGAGCACTGATGT
	R: GGGATCTCTGTCTCCAGCACCA
Notch1	F: AGAGGAACTCAAGGGGAGCG
	R: CGGCACTCGCATTTGTAGTC
Notch2	F: CACACCCGAGTGCTTGTTTG
	R: ATTACAGCCCTGGTCACAGC
JAG1	F: AACTGGTACCGCTGTGAGTG
	R: GCAAGGCCTCCCTGTAACTT
JAG2	F: CAAGTGGCTGGGAAGGAGAA
	R: TGCATCGGCCACCATTATGA
DLL1	F: TGAACTACTGCACTCACCACAA
	R: TCGTTGATTTCAATCTCGCAGC
DLL4	F: CAGCAGGTAACGGTCGGAG
	R: TTGACGAACTCGTGCAGCTT
RBPJ	F: GCAGATGATCCGGTATCGCAG
	R: TTTGGGCATGGAGTGGCTTGA
HES1	F: CAGCGAGTGCATGAACGAAG
	R: TGATGGCGTTGATCTGGGTC
HES4	F: CCCATCATGGAGAAGCGACG
	R: GAGTGCCGAGAGCTGTCTTTT
HES5	F: TGAAATACAGCCGAGCTTTTGC
	R: GCAGAAGGAGAACCGGAGTC
HES6	F: CGCATTCCACTTGGATCAGTCTA
	R: AGGCCCACTTTGGAATCAGC
HEY1	F: GGCCGGAGGGAAAGGTTATTT
	R: GTGATGTCCAAAGGCGTTGC
HEY2	F: TATTTCTCTTTGCCCCACGCC
	R: TATGGCTTTGCCCGCAGTA
HEYL	F: TCAGGATGAAGCGTCTGTGC R: GCCGCTTCTCAATGATCCCT
Caspase-3	F: TGCTCCAGGCTACTACTCC R: CCACTCTGCGATTTACACGA
Caspase-9	F: CGAAGGAGCAAGCACGAC R: CGCAGCCCTCATCTAGCAT
Bcl-2	F: CGACTGGGATGACAGGAAAG R: GGAGCGCACAGGTGAGACA
GAPDH	F: GAACATCATCCCAGCGTCCA
	R: CGGCAGGTCAGGTCAACAAC

## Data Availability

No new data were created or analyzed in this study. Data sharing is not applicable to this article.

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
