# Peer review of "Notch-RBPJ Pathway for the Differentiation of Bone Marrow Mesenchymal Stem Cells in Femoral Head Necrosis"

_ijms, 2023, doi:10.3390/ijms24076295_

Round 1

Reviewer 1 Report

The authors modeled necrosis using MP and investigated how the Notch-RBPJ pathway regulates MSC differentiation. Here are some questions:

·         Do the human MSCs respond the same to MP treatment or Notch-RBPJ inhibitor/activator?

·         How does the Notch pathway regulate intramembranous/endochondral ossification?

Author Response

Dear reviewer

Thank you very much for your comments. We have revised the manuscript carefully and would like to re-submit it for your consideration. The amendments were emphasized in yellow color in the revised manuscript. Please see the attachment and we hope that the revision is acceptable.

Yours sincerely

Zhenlei Zhou

College of Veterinary Medicine

Nanjing Agricultural University

Nanjing, China

Tel: +86 025 84395505

E-mail: zhouzl@njau.edu.cn

Reviewer 2 Report

The authors are addressing femoral head necrosis in broilers, where bone marrow mesenchymal cells show enhanced lipogenic differentiation and diminished chondrogenic and osteogenic differentiation. The authors show that Notch signaling related factors are up regulated. But the presentation quality needs serious revision.

1. In Fig 1B, the Alcian blue staining image for the control and the FHN looks exactly the same and is highly pixelated. Are they duplicate images of the same field? Then please be careful and replace that.

2. The Western Blots in Fig2G and 2I needs to be repeated, atleast the panels for BMP2, MMP9 and Notch2. They don’t look neat.

3. The control image for immunofluorescence in Fig2J, especially for RBPJ and merge looks very odd. What is the black gap in the middle of the figure? Please replace that with a better image.

4. In fig 5J, the si-RBPJ addition seems to not diminish RBPJ expression in the nucleus, what are the author’s thoughts on this? Should they use a different si-RNA or shRNA approach?

5. Please rewrite line 91-92.

Author Response

(The authors gave the same response as above.)

Round 2

Reviewer 1 Report

The authors have addressed all of my concerns.

Reviewer 2 Report

The authors addressed sincerely all the comments and questions I had. Thank you.